# Association of Red Blood Cell Life Span with Abnormal Changes in Cardiac Structure and Function in Non-Dialysis Patients with Chronic Kidney Disease Stages 3–5

**DOI:** 10.3390/jcm11247373

**Published:** 2022-12-12

**Authors:** Siyi Rao, Jing Zhang, Jiaqun Lin, Jianxin Wan, Yi Chen

**Affiliations:** 1Department of Nephrology, Blood Purification Research Center, The First Affiliated Hospital, Fujian Medical University, Fuzhou 350005, China; 2Fujian Clinical Research Center for Metabolic Chronic Kidney Disease, The First Affiliated Hospital, Fujian Medical University, Fuzhou 350005, China; 3Department of Nephrology, National Regional Medical Center, Binhai Campus of the First Affiliated Hospital, Fujian Medical University, Fuzhou 350212, China

**Keywords:** red blood cell life span, cardiac structure and function, chronic kidney disease, heart failure

## Abstract

Introduction: With the invention and improvement of the carbon monoxide (CO) breath test, the role of shortened red blood cell life span (RBCLS) in renal anemia, an independent risk factor for cardiovascular events in patients with chronic kidney disease (CKD), is gradually attracting attention. Considering that heart failure is the leading cause of morbidity and mortality in patients with CKD, this study investigated the correlation between the RBCLS and the cardiac structure and function in non-dialysis patients with CKD stages 3–5, aiming to provide new ideas to improve the long-term prognosis of CKD patients. Methods: One hundred thirty-three non-dialysis patients with CKD stages 3–5 were tested for RBCLS. We compared the serological data, cardiac ultrasound results, and follow-up prognosis of patients with different RBCLS. Results: As the RBCLS shortened, the patients’ blood pressure, BNP, and CRP gradually increased, most significantly in patients with an RBCLS < 50 d. Patients with an RBCLS < 50 d had substantially lower hemoglobin (Hb), hematocrit, and albumin levels than those with an RBCLS ≥ 50 d. The cardiac ultrasound results show that patients with an RBCLS < 50 d had significantly larger atrial diameters than those with an RBCLS ≥ 50 d and were associated with more severe diastolic dysfunction. Patients with an RBCLS < 50 d had a 3.06 times greater risk of combined heart failure at baseline than those with an RBCLS ≥ 70 d and a higher risk of heart failure at follow-up. CKD stage 5 patients with an RBCLS < 50 d were more likely to develop heart failure and require renal replacement therapy earlier than patients with an RBCLS ≥ 50 d. Conclusions: In non-dialysis patients with CKD stages 3–5, there is a correlation between the red blood cell life span and cardiac structure and function. The RBCLS may also impact the renal prognosis of CKD patients.

## 1. Introduction

Chronic kidney disease (CKD) has become a significant public health threat worldwide, with high morbidity and mortality rates. The global prevalence of CKD is estimated to be 13.4%, with a plurality of 10.6% for CKD stages 3–5 [1]. This group of patients is more likely to progress to end-stage renal disease (ESRD) and is at higher risk of cardiovascular events. CKD is associated with an increased risk of cardiovascular disease [2]. This risk increases with the progression of chronic kidney disease [3].

Traditional risk factors for cardiovascular events include smoking, diabetes, hypertension, and hyperlipidemia [4]. In recent years, studies have suggested that renal anemia is also an independent risk factor for cardiovascular events in patients with CKD [5]. Anemia can increase the cardiac load. Severe anemia can lead to decreased myocardial contractility, reduced ejection fraction, and heart failure, leading to an increased progression of CKD and anemia, creating a vicious circle of the three.

The leading cause of renal anemia is the reduction in erythropoietin (EPO). Other causes include erythrocyte malformations due to the uremic environment, folic acid and vitamin B12 deficiency, dysregulated iron homeostasis, and blood loss during hemodialysis. Recently, with the invention and improvement of the CO breath test, the issue of shortened RBCLS has attracted attention. Jiuhong-Li [6] et al. determined by the Levitt CO breath test that the erythrocyte life spans of patients with CKD1-5 were 122 ± 50, 112 ± 26, 90 ± 32, 88 ± 28, and 60 ± 24 days, respectively, and concluded that the erythrocyte life span of patients with CKD3-5 was significantly shorter than that of patients with CKD1-2. Is there an association between a shortened RBCLS and cardiovascular events in patients with CKD? There is a lack of studies in this area. In this study, we used the CO breath test to evaluate the RBCLS of patients with CKD stages 3–5 and the correlation between the erythrocyte life span and cardiac function in CKD patients, providing new ideas to improve the long-term prognosis of CKD patients.

## 2. Materials and Methods

### 2.1. Study Subjects

A cohort of 133 patients diagnosed with CKD was recruited from the Department of Nephrology at the First Affiliated Hospital, Fujian Medical University, Fuzhou, China. According to the condition of RBCLS, we divided the patients into four groups: 54 cases with normal RBCLS (RBCLS ≥ 70 days), 28 cases with mild shortening (60 days ≤ RBCLS < 70 days), 31 cases with moderate shortening (50 days ≤ RBCLS < 60 days), and 20 cases with severe shortening (RBCLS < 50 days). We required smokers to abstain from smoking within 24 h prior to RBCLS testing.

We excluded patients with a follow-up time of fewer than six months;Oral roxadustat (FG-4592) for more than one month;Other causes of anemia (based on clinical presentation, past medical history, family history of disease, and serological data);A clear sign of acute infection;Unable to cooperate with the test due to a combination of chronic lung disease or cancer;Congenital heart disease and a previous diagnosis of coronary artery disease or atrial fibrillation.

All of the patients provided written informed consent to participate in the study. The Institutional Review Board of First Affiliated Hospital, Fujian Medical University approved the study protocol.

### 2.2. Indicators of Anemia

On the same day that alveolar air sampling was performed, we collected peripheral venous blood samples to detect anemia indicators, including Hb, hematocrit, and red cell volume distribution width (RDW).

### 2.3. RBCLS

There are two primary sources of CO in human alveolar exhaled gas: endogenous metabolism and exogenous inhalation. About 86% of the endogenous CO comes from the degradation of hemoglobin, and about 85% of the CO derived from the degradation of hemoglobin comes from the degradation of hemoglobin in erythrocytes. Therefore, about 70% of the endogenous CO comes from the degradation of hemoglobin after the destruction of red blood cells, and the lung is the only method of CO excretion. In 2018, Professor Yongjian-Ma [7] et al. developed the world’s first red blood cell life span meter, which is based on the principle of non-dispersive infrared spectroscopy for the detection of carbon monoxide in breath traces, making the instrument easier to operate and the results very reliable. The alveolar and dead airbags are 1500 mL and 300 mL, respectively, and made of aluminum foil. The patient is asked to sit still, breathe deeply, and then hold the breath for 10 s. The cavity airbag is first filled when blowing. This portion of the gas is the physiological dead cavity air and is discarded, and the subsequent exhalation enters the alveolar airbag. Finally, the Levitt formula is used to calculate the erythrocyte life span. Each patient’s red blood cell life span was detected by our hospital ECT room.

### 2.4. Cardiac Color Doppler Index

Patients were placed in a supine position, and all data were recorded with a two-dimensional color ultrasound: left atrial internal diameter (LAD), left ventricular end-diastolic internal diameter (LVDd), left ventricular end-systolic internal diameter (LVDs), interventricular septum thickness (IVST), thickness of the left ventricular posterior wall (LVPW), left ventricular ejection fraction (LVEF), early diastolic blood flow velocity at the mitral cusp/early diastolic myocardial motion velocity at the mitral annulus (E/e), left ventricular myocardial mass (LVM), and cardiac output (CO).

### 2.5. Definition of Heart Failure

We excluded patients with a BNP level < 100 pg/mL before diagnosing heart failure. Heart failure was diagnosed according to the definition of heart failure mentioned in the report “Universal definition and classification of heart failure”, published in 2021 [8]. Heart failure was diagnosed by combining cardiac structure and function (as determined by an EF < 50%, abnormal cardiac chamber enlargement, E/E′ > 15, moderate/severe ventricular hypertrophy or moderate/severe valvular obstructive or regurgitant lesion), and clinical symptoms and signs [8].

Left ventricular hypertrophy (LVH) was diagnosed by the left ventricular myocardial mass index (LVMI). An LVMI > 130 g/m^2^ in men and an LVMI > 110 g/m^2^ in women were considered to indicate the presence of LVH [9]. Abnormal chamber enlargement was defined as an enlarged left atrium (LAD > 3.8 cm women, LAD > 4.0 cm men) or enlarged left ventricle diastolic dimension (LVDd > 4.86 cm women, LVDd > 5.53 cm men) [10].

### 2.6. Definition of Calcified Heart Valves

In this study, echocardiography was used to assess the presence of strong echogenic hard bands in the aortic and mitral valves, the presence of either of which was diagnosed as cardiac valve calcification [11].

### 2.7. Follow-Up Time and Indicators

We used the outpatient medical record system to access the patients’ follow-up records as well as telephone follow-ups. The follow-up indicators included the following:Whether or not they survived;Whether or not they entered renal replacement therapy (hemodialysis, peritoneal dialysis, renal transplantation), and the time to enter renal replacement therapy;According to a 2020 Chinese Hypertension Survey [12] on heart failure and left ventricular insufficiency in elderly patients with chronic kidney disease, we investigated 4 typical symptoms, including:Reduced ability to exercise;Shortness of breath (dyspnea) with exertion;Swelling (edema) in the legs, ankles, and feet;Shortness of breath (dyspnea) when lying down.

In a sensitivity analysis, at least three out of four typical HF symptoms were used to define HF [12]. Therefore, in this study, the presence of three or more typical symptoms at the time of follow-up was considered heart failure.

### 2.8. Statistical Methods

The data were processed and statistically analyzed using STATA.16 software (Stata Corp., College Station, TX, USA). The quantitative data were presented as the mean ± standard deviation (SD) or median with the interquartile range (IQR) depending on whether the data were normally distributed. Linear regression equations were used to explore the relationship between the RBCLS and the cardiac ultrasound indices. Relative risk regression with Poisson error distributions and robust standard errors were used to estimate the association between the RBCLS and the prevalence of heart failure. Correlation models were used to adjust for confounders and to examine whether these potential factors changed the strength of the association between the RBCLS and the prevalence of heart failure. The Kaplan–Meier method was used to compare the time to renal replacement therapy and the time to the onset of heart failure symptoms, and the log-rank test was used to compare the groups. A *p* < 0.05 was considered to be a statistically significant difference.

## 3. Results

### 3.1. Baseline Characteristics of the Study Cohort

A total of 133 participants were included in the overall sample; 78 (58.65%) were male, the mean age was 58.1 ± 15.7 years old, and the median follow-up was eight months (interquartile range, 4–13 months). The three most common causes of CKD included chronic GN, DM, and hypertension. A total of 10 (7.52%) had stage 3a CKD, 23 (17.29%) had stage 3b CKD, 26 (19.55%) had stage 4 CKD, and 74 (55.64%) had stage 5 CKD. The median RBCLS among the 133 participants was 64 days, and the mean RBCLS value from CKD3b to CKD5 was less than 70 days (Figure 1). The RBCLS was inconsistently related to traditional cardiovascular risk factors (Table 1). On one hand, lower RBCLS values were associated with higher blood pressure, higher BNP, CRP, and red blood cell distribution width. On the other hand, lower RBCLS values were associated with the male gender, lower hemoglobin, and lower hematocrit. Moreover, participants with an RBCLS < 50 days had lower albumin and serum ferritin. No significant differences were seen in the calcium, phosphorus, or parathyroid hormone (PTH) levels among the four groups. Additionally, there were no significant differences among the groups in terms of cardiac or anemia treatment.

### 3.2. RBCLS and Baseline Echocardiographic Index

At the baseline data collection, 124 patients had simultaneous color Doppler echocardiography. Patients with an RBCLS of fewer than 50 days appeared to have larger hearts (Table 2). A lower RBCLS was associated with higher LAD. The mean values of the LVDs, LVPW, LVM, and E/e were greater in the RBCLS < 50 days group than in the other three groups. However, there were no significant differences among the four groups in the LVDd, IVST, or LVEF. Of note, the RBCLS was positively correlated with the LVEF (*p* = 0.025) and negatively correlated with the LAD (*p* = 0.023), LVDs (*p* = 0.002), LVDd (*p* = 0.013), LVPWT (*p* = 0.007), LVM (*p* < 0.001), and E/e (*p* = 0.012), respectively (Figure 2).

### 3.3. Association between RBCLS and Heart Failure

We observed an association between the RBCLS and the prevalence of heart failure at baseline in non-dialysis patients with CKD stages 3–5 (Table 3). After adjusting for common risk factors for heart failure, such as gender, age, smoking status, the combination of hypertension and diabetes, and heart valve calcification conditions, the prevalence of heart failure was 3.06 times higher in the RBCLS < 50 d group than in the RBCLS ≥ 70 d group. Adjustments for the treatment status, serum creatinine, hemoglobin, CRP, RDW, and serum albumin did not alter the strength of any of the prevalence ratio (PR) estimates (Table 3, Models 2 and 3).

The patients’ symptoms were investigated by telephone to ensure that each patient was followed up for more than six months. The occurrence of heart failure during the follow-up period was defined by the number of typical and atypical heart failure symptoms. Our follow-up results suggest that the prevalence of heart failure tended to increase as the RBCLS shortened (Table 4).

### 3.4. RBCLS and Time to Renal Replacement

During the follow-up of patients with CKD stage 5 (median follow-up period of 8 months), 56 (75.68%) patients entered dialysis; of these, 45 (60.81%) patients were on hemodialysis, 10 (13.51%) started peritoneal dialysis, and 1 underwent a kidney transplant. The Kaplan–Meier analysis revealed that during the 12 months of follow-up, patients in the RBCLS < 50 d group required renal replacement therapy earlier than those in the RBCLS ≥ 50 d group (Figure 3A) (*p* = 0.016).

### 3.5. RBCLS and the Time at Which Heart Failure Occurs

Patients were followed up by telephone for symptoms of heart failure. A total of 16 (21.62%) patients developed heart failure during follow-up. The Kaplan–Meier analysis showed that during the 12 months of follow-up, the patients with an RBCLS < 50 days were more likely to develop heart failure than those with an RBCLS ≥ 50 days (Figure 3B) (*p* = 0.037).

## 4. Discussion

In this study of non-dialyzed chronic kidney disease patients with CKD3-5, we found that a reduction in the red blood cell life was not only associated with the degree of anemia and nutritional status of the patients but also associated with the presence or absence of cardiac structural and functional abnormalities. To our knowledge, this is the first study to link red blood cell longevity to cardiac function in patients with CKD.

The RBCLS refers to the time red blood cells survive in circulation after being released from the bone marrow. The average life span of red blood cells in healthy people is about 120 days. The shortening of the RBCLS is a reliable marker of the accelerated destruction of red blood cells. Based on the standard marker method, the range of the RBCLS in healthy individuals is 70–140 days [13,14,15]. In our study, the RBCLS was less than the minimum normal value for 70 days at all stages when CKD progressed to stages 3b–5. Hou-de Zhang [7] et al. recently used Levitt’s CO breath test to compare the RBCLS values of healthy people with those of hemolytic patients and found that the 75-day cutoff was 100% accurate in diagnosing hemolytic anemia. It can be inferred that most patients with CKD stages 3b-5 have a high likelihood of combined hemolysis.

Erythrocytes are the carriers of oxygen and carbon dioxide exchange between the lungs and peripheral tissues. However, the link between erythrocytes and cardiovascular events has received increasing attention in recent years with the discovery that erythrocytes are also involved in endothelial cell damage in diabetes and thus mediate early cardiovascular events in diabetic patients [16]. Evidence suggests that erythrocytes play an important role in the regulation of cardiovascular homeostasis through the expression of biologically active substances, such as adenosine triphosphate (ATP) and nitric oxide (NO) [17,18], as well as through their well-developed antioxidant system. Erythrocytes may undergo “erythropathy” in various disease states, with the main manifestations including an increased formation of reactive oxygen species (ROS) [17,19], altered protein content and enzymatic activity [20,21,22], and increased adhesion to the vessel wall [23].

Patients with CKD often have a combination of internal environmental factors, such as an increased expression of inflammatory factors, oxidative stress, uremic toxins, and disturbances in calcium and phosphorus metabolism, all of which impact erythrocyte homeostasis and mediate erythrocyte pathology. Erythrocytes contain superoxide dismutase and glutathione peroxidase, which are efficient free radical scavengers that protect the cell membrane from damage by oxygen radicals.

CKD is associated with increased cardiovascular mortality and morbidity [24]. Several studies have shown that CKD is still associated with cardiovascular mortality after adjusting for cardiovascular disease risk factors, including diabetes, hypertension, obesity, dyslipidemia, and hyperuricemia [25,26]. This suggests that multiple factors other than traditional cardiovascular factors are involved in cardiovascular mortality in patients with CKD. Hickson [27] et al. found that the left ventricular ejection fraction (LVEF) and the right ventricular systolic dysfunction (RV) were associated with a median follow-up time of 2.4 years for death by analyzing the cardiac ultrasounds of patients 1–3 months prior to dialysis initiation, after adjusting for age and gender. In a single-center study by Pluta [28] et al., the risk was also recently found to be associated with LV remodeling at an early stage of the development of chronic kidney disease. In contrast, for the first time in our study, a link was constructed between the erythrocyte life span and the cardiovascular structure and function in CKD patients. In our study, we found that as the erythrocyte life span shortened, the atrial internal diameter of the heart became longer and the cardiac mass increased in CKD patients. Nevertheless, both systolic and diastolic functions were reduced. This suggests that the erythrocyte life span is associated with structural and functional abnormalities in the hearts of CKD patients.

In this study, patients in the RBCLS < 50 group had a heart failure prevalence 3.06 times higher than that of patients in the RBCLS ≥ 70 group. We also found that the symptoms of heart failure became more pronounced as the erythrocyte life span decreased during the follow-up of patients with symptoms. This suggests that there may also be an association between the red blood cell life span and the occurrence of cardiovascular events. However, the underlying mechanisms are not fully understood. Considering that a shortened RBCLS is an essential factor in the development and progression of renal anemia, anemia is a significant risk factor for cardiovascular events in CKD patients. The RBCLS may further influence the cardiac load by affecting the hemoglobin content, causing abnormalities in the cardiac structure and function.

Another explanation may be based on the hypothesis of arginase activation. Patients with CKD have a disturbed internal environment, with increased production of reactive oxygen species (ROS) due to erythrocyte lesions caused by various factors, such as inflammatory factors, oxidative stress, and uremic toxins. Arginase hydrolyses L-arginine to ornithine and urea, which are critical regulatory mechanisms for NO production in endothelial cells [29]. The conversion of L-arginine to ornithine and urea inhibits the utilization of the substrate (L-arginine), leading to a decrease in the total nitric oxide (NO) in patients. The decrease in the total NO leads to vascular endothelial cell dysfunction and impaired vasodilatation, further mediating the occurrence of cardiovascular events. Studies have shown that arginase inhibition has improved microvascular and macrovascular endothelium in patients with type 2 diabetes [21,30]. Yang J. et al. [31] demonstrated that the inhibition of RBC arginase significantly improved cardiac recovery after ischemia-reperfusion.

The incidence of heart failure in patients with CKD ranges from approximately 17% to 21% [32] and is one of the leading causes of emergency department visits and hospital admissions [33], which means a poor health-related quality of life [34]. Patients with CKD usually present first with diastolic heart failure, i.e., ejection fraction preserved heart failure [35]. We have considered whether cardiac structure and function abnormalities act in opposite directions to cause a shortening of the RBCLS. Patients with heart failure have reduced tissue perfusion, increased oxidative stress, and reduced oxygen supply, causing impaired microcirculation and endothelial cell function. In this context, endothelial cells express increased adhesion molecules, pro-inflammatory factors, and endothelial reactive oxygen species, causing an altered rheological state of hypercoagulable blood [36]. This set of changes may cause erythropathy and accelerate the reticuloendothelial system’s premature clearance of defective red blood cells, resulting in a shortened RBCLS.

Several risk factors influence CKD patients’ prognosis, including age, diabetes, hypertension, hyperlipidemia, and obesity. In recent years, it was also found that the left ventricular mass index (LVMI), proteinuria, and hemoglobin levels are independently associated with dialysis progression in patients with CKD [37]. Therefore, the treatment of anemia, proteinuria, and hypertension, and the prevention of LV hypertrophy due to long-term volume overload is essential to delay the progression of dialysis in CKD patients. Considering the small number of patients with CKD stages 3–4 included in this study, a Kaplan–Meier survival curve analysis was performed for patients with CKD stage 5 only, and it was found that patients with an RBCLS < 50 days in CKD stage 5 required renal replacement therapy and presented with heart failure earlier than those with an RBCLS > 50 days. The RBCLS may also be a factor in the prognosis of CKD patients.

The limitations of this study are also worth mentioning, starting with the fact that the single-center survey limits the generalizability of our findings. Despite the extensive adjustments, residual confounding is still possible. The assessment of cardiac function during follow-up was based only on the patients’ subjective perceptions and lacked objective indicators. In addition, the diagnosis of heart failure in this study was based on a universal diagnostic criterion, which may have biased the diagnosis of heart failure in patients with CKD. Additionally, we used the index LAD/LVDd, which may not be as accurate as the left atrial/left ventricular volume index.

In conclusion, this study found an association between the erythrocyte life span and abnormalities in the cardiac structure and function in non-dialysis patients during CKD3-5. The shortened erythrocyte life span may suggest structural changes in the heart, with potential implications for the occurrence of long-term cardiovascular events and renal prognosis. However, a large, multi-center study with a prospective design is needed to further confirm the correlation, and the underlying pathophysiological mechanisms need further exploration.

## 5. Conclusions

In non-dialysis patients with CKD stages 3–5, there is an association between the red blood cell life span and cardiac structure and function. Patients with shorter red blood cell life spans are more likely to develop cardiac structure and function. There may also be implications for the long-term prognosis of the kidney and the occurrence of cardiovascular events in CKD patients.

## Figures and Tables

**Figure 1 jcm-11-07373-f001:**
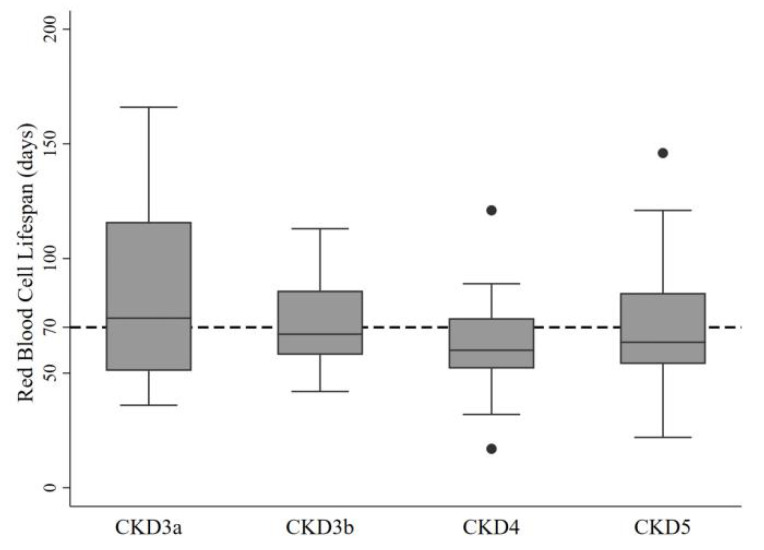
Distribution of RBCLS in patients with CKD stages 3–5.

**Figure 2 jcm-11-07373-f002:**
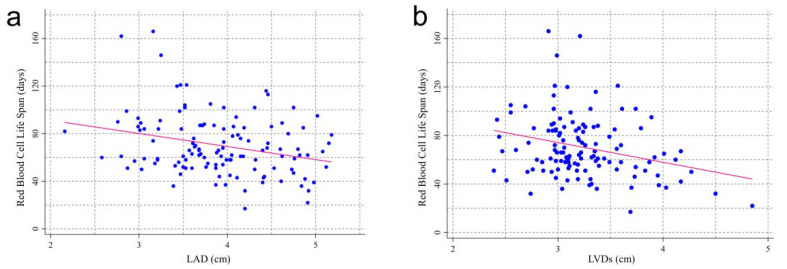
Relationship between RBCLS and cardiac color Doppler ultrasound indicators. Each data point is an individual measurement. The pink line represents the linear regression fitting line. (**a**) LAD (beta = −10.98, *p* = 0.02); (**b**) LVDs (beta = −16.30, *p* = 0.002); (**c**) LVDd (beta = −10.82, *p* = 0.013); (**d**) IVST (beta = −21.12, *p* = 0.061); (**e**) LVPWT (beta = −36.03, *p* = 0.007); (**f**) LVM (beta = −0.12, *p* = 0.001); (**g**) LVEF (beta = 0.87, *p* = 0.025); (**h**) E/e (beta = −1.28, *p* = 0.012).

**Figure 3 jcm-11-07373-f003:**
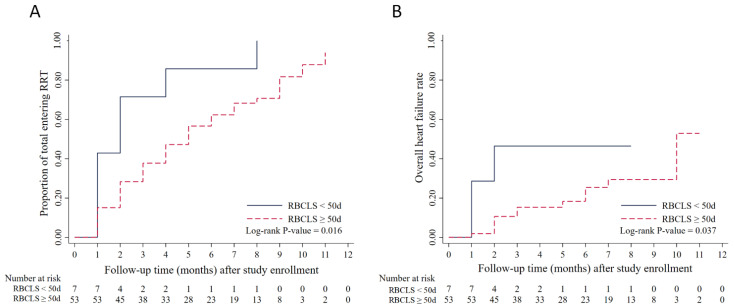
Kaplan–Meier curves for time to renal replacement therapy and heart failure in patients with different RBCLS. (**A**) Kaplan–Meier curves for time to renal replacement therapy in patients with different RBCLS. (**B**) Kaplan–Meier curves for time to heart failure in patients with different RBCLS.

**Table 1 jcm-11-07373-t001:** Comparison of general data in non-dialysis patients with CKD stages 3–5 with shortened RBCLS.

Characteristic	Red Blood Cell Life Span (Day)				
	≥70	60 to 70	50 to 60	<50	*p*
	(*n* = 54)	(*n* = 28)	(*n* = 31)	(*n* = 20)	
Demographics					
Age (mean (SD))	60.6 (16.0)	53.9 (16.3)	55.7 (13.4)	61.0 (17.0)	0.195
Male gender (*n* (%))	32 (59.3)	18 (64.3)	12 (38.7)	16 (80.0) ^c^	0.027
Cardiovascular risk factors					
Smoking (*n* (%))	6 (11.1)	5 (17.9)	3 (9.7)	5 (25.0)	0.353
BMI (mean (SD))	23.5 (3.1)	22.5 (3.0)	23.8 (3.6)	23.3 (2.6)	0.385
Diabetes (*n* (%))	23 (42.6)	8 (28.6)	11 (35.5)	9 (45.0)	0.564
Hypertension (*n* (%))	44 (81.5)	25 (89.3)	27 (87.1)	20 (100)	0.186
Anemia (*n* (%))	49 (91.8)	26 (92.3)	31 (100)	20 (100)	0.203
SBP (mmHg; mean (SD))	135.6 (22.4)	138.7 (27.4)	141.6 (27.8)	155.9 (24.1) ^a^	0.023
DBP (mmHg; mean (SD))	75.0 (14.0)	79.8 (17.0)	85.0 (16.7) ^a^	84.1 (20.1)	0.031
MAP (mmHg; mean (SD))	95.2 (14.0)	99.4 (18.5)	103.8 (18.9)	108.0 (18.1) ^a^	0.017
LDL cholesterol (mmol/L; mean (1/4, 3/4))	2.4 (1.8, 3.0)	2.8 (2.4, 3.3)	2.3 (2.0, 2.9)	2.6 (2.4, 3.6)	0.067
Triglycerides (mmol/L; mean (1/4, 3/4))	1.5 (0.9, 2.2)	1.4 (1.1, 2.0)	1.6 (1.1, 2.0)	1.4 (1.0, 2.0)	0.948
C-reactive protein (mg/L; mean (1/4, 3/4))	5.0 (5.0, 6.0)	5.0 (5.0, 9.5)	5.0 (5.0, 8.6)	9.4 ^abc^ (5.0, 25.5)	0.027
BNP (ng/L; mean (1/4, 3/4))	48.0 (14.0, 224.6)	80.0 (33.0, 244.0)	65.5 (30.6, 252.3)	330.7 ^abc^ (166.0, 518.7)	0.003
eGFR (mL/min per 1.73 m^2^; mean (1/4, 3/4))	12.8 (9.5, 34.5)	17.6 (9.2, 35.6)	10.5 (5.8, 18.7)	16.4 (7.1, 33.6)	0.221
Other serological indicators					
Hemoglobin (g/L; mean (SD))	92.0 (19.3)	96.0 (21.7)	83.6 (11.7) ^ab^	79.4 (19.7) ^ab^	0.004
Hematocrit (L/L; mean (SD))	0.28 (0.06)	0.29 (0.06)	0.25 (0.04) ^ab^	0.24 (0.06) ^ab^	0.005
RDW (%; mean (1/4, 3/4))	15.4 (13.7, 15.4)	14.5 (13.5, 16.0)	14.5 ^ab^ (13.3, 15.6)	15.8 ^ab^ (15.0, 17.0)	0.032
Serum albumin (g/L; mean (1/4, 3/4))	37.0 (32.2, 29.9)	36.4 (31.7, 39.4)	32.7 ^a^ (29.4, 38.3)	32.9 ^ab^ (28.5, 35.2)	0.020
Transferrin saturation (%; mean (1/4, 3/4))	22.6 (13.23, 32.77)	20.1 (14.96, 30.99)	31.33 (19.63, 46.06)	20.95 (13.58, 29.37)	0.1154
Corrected calcium (mmol/L; mean (1/4, 3/4))	2.2 (2.2, 2.3)	2.3 (2.2, 2.4)	2.2 (2.1, 2.3)	2.2 (2.1, 2.3)	0.480
Phosphorus (mmol/L; mean (1/4, 3/4))	1.4 (1.1, 1.8)	1.4 (1.2, 1.6)	1.6 (1.3, 1.8)	1.5 (1.1, 1.9)	0.618
Parathyroid hormone (pmol/L; mean (1/4, 3/4))	62.5 (39.0, 99.0)	77.0 (42.5, 99.5)	57.0 (26.0, 92.0)	84.0 (26.5, 104.5)	0.681
Medication status					
EPO (*n* (%))	21 (38.89)	11 (39.29)	17 (54.84)	9 (45.00)	0.511
Iron (*n* (%))	12 (22.22)	3 (10.71)	6 (19.35)	3 (15.00)	0.637
ACEI/ARB (*n* (%))	9 (16.67)	2 (7.14)	2 (6.45)	4 (20.00)	0.332
β-blockers (*n* (%))	23 (42.59)	16 (57.14)	15 (48.39)	11 (55.00)	0.589
α-blockers (*n* (%))	14 (25.93)	7 (25.00)	11 (35.48)	10 (50.00)	0.194
Calcium channel blocker (*n* (%))	42 (77.78)	24 (85.71)	27 (87.10)	19 (95.00)	0.299
Spironolactone (*n* (%))	3 (5.56)	4 (14.29)	3 (9.68)	4 (20.00)	0.243

BMI, body mass index; DBP, diastolic BP; SBP, systolic BP; MAP, mean arterial pressure; BNP, brain natriuretic peptide; RDW, red blood cell distribution width; ACEI, angiotensin-converting enzyme inhibitors; ARB, angiotensin receptor blockers; ^a^
*p* < 0.05 vs. RBCLS ≥ 70 d group; ^b^
*p* < 0.05 vs. 60 ≤ RBCLS < 70 d group; ^c^
*p* < 0.05 vs. 50 ≤ RBCLS < 60 d group.

**Table 2 jcm-11-07373-t002:** Echocardiographic features of non-dialysis patients with CKD stages 3–5 with shortened RBCLS.

Characteristic	Red Blood Cell Life Span (Day)				
	≥70	60 to 70	50 to 60	<50	*p*
	(*n* = 52)	(*n* = 26)	(*n* = 27)	(*n* = 19)	
Cardiac ultrasound indicators					
LAD (cm; mean (SD))	3.78 (0.67)	4.00 (0.61)	3.76 (0.59)	4.33 (0.49) ^ac^	0.006
LVDd (cm; mean (SD))	4.85 (0.46)	4.97 (0.50)	5.04 (0.52)	5.20 (0.65)	0.075
LVDs (cm; mean (1/4, 3/4))	3.11 (2.96, 3.29)	3.16 (3.03, 3.38)	3.16 (2.98, 3.56)	3.37 ^abc^ (3.09, 3.96)	0.032
IVST (cm; mean (1/4, 3/4))	1.06 (0.89, 1.15)	1.05 (0.93, 1.19)	1.01 (0.87, 1.14)	1.14 (1.05, 1.21)	0.079
LVPW (cm; mean (1/4, 3/4))	0.95 (0.87, 1.04)	0.98 (0.86, 1.11)	0.90 (0.78, 1.00)	1.05 ^abc^ (0.96, 1.21)	0.003
LVEF (%; mean (1/4, 3/4))	64.57 (61.76, 67.66)	64.54 (61.17, 66.19)	64.58 (62.00, 67.65)	60.00 (54.91, 64.15)	0.058
E/e (%; mean (1/4, 3/4))	9.43 (7.52, 12.72)	9.06 (8.00, 12.67)	10.00 (8.00, 14.57)	14.00 ^abc^ (10.60, 15.23)	0.038
LVM (g; mean (1/4, 3/4))	169.0 (132.4, 212.6)	180.9 (140.0, 221.0)	174.2 (145.2, 199.3)	226.7 ^abc^ (169.0, 308.0)	0.018
CO (L/min; mean (1/4, 3/4))	5.09 (4.35, 6.00)	5.39 (4.59, 6.56)	5.86 (5.13, 7.02)	5.00 (4.01, 8.00)	0.082
Calcified heart valves (*n* (%))	9 (17.31)	6 (22.22)	9 (33.33)	5 (26.32)	0.442

LAD, left atrial internal diameter; LVDd, left ventricular end-diastolic internal diameter; LVDs, left ventricular end-systolic internal diameter; IVST, interventricular septum thickness; LVPW, thickness of left ventricular posterior wall; LVEF, left ventricular ejection fraction; E/e, early diastolic blood flow velocity at the mitral cusp/early diastolic myocardial motion velocity at the mitral annulus; LVM, left ventricular myocardial mass; CO, cardiac output. ^a^
*p* < 0.05 vs. RBCLS ≥ 70 d group; ^b^
*p* < 0.05 vs. 60 ≤ RBCLS < 70 d group; ^c^
*p* < 0.05 vs. 50 ≤ RBCLS < 60 d group.

**Table 3 jcm-11-07373-t003:** Association analysis of different RBCLS with heart failure at baseline.

Red Blood Cell Life Span (Days)	Prevalence of Heart Failure (%)	Model 1		Model 2		Model 3	
		PR	95% CI	PR	95% CI	PR	95% CI
RBCLS ≥ 70	25.93	1.00	Reference	1.00	Reference	1.00	Reference
60 ≤ RBCLS < 70	32.14	1.48	0.61, 3.59	1.23	0.52, 2.90	1.19	0.51, 2.80
50 ≤ RBCLS < 60	35.48	1.47	0.62, 3.49	1.24	0.56, 2.77	1.14	0.49, 2.65
RBCLS < 50	80.00	3.06	1.41, 6.64 *	2.68	1.27, 5.66 *	2.27	1.02, 5.06 *
*p*		0.005		0.010		0.045	

CI, confidence interval. Model 1 adjusted for sex, age, smoking, hypertension, diabetes, and calcified heart valves. Model 2 adjusted for EPO, iron, ACEI/ARB, β-blockers, α-blockers, calcium channel blockers, and spironolactone therapy. Model 3 adjusted for serum creatinine, hemoglobin, CRP, RDW, and albumin. * *p* < 0.05 vs. RBCLS ≥ 70 d group. *p* for trend > 0.05.

**Table 4 jcm-11-07373-t004:** Association analysis of different RBCLS rates with heart failure at the end of follow-up.

Red Blood Cell Life Span (Days)	Prevalence of Heart Failure (%)	Model 1		Model 2		Model 3	
		PR	95% CI	PR	95% CI	PR	95% CI
RBCLS ≥ 70	16.67	1.00	Reference	1.00	Reference	1.00	Reference
60 ≤ RBCLS < 70	17.86	0.89	0.28, 2.78	1.04	0.34, 3.21	0.91	0.30, 2.82
50 ≤ RBCLS < 60	22.58	1.42	0.49, 4.07	1.19	0.44, 3.26	1.19	0.42, 3.42
RBCLS < 50	30.00	1.29	0.44, 3.77	1.37	0.45, 4.17	1.80	0.57, 5.70
*p*		0.642		0.583		0.456	

CI, confidence interval. Model 1 adjusted for sex, age, smoking, hypertension, diabetes, and calcified heart valves. Model 2 adjusted for EPO, iron, ACEI/ARB, β-blockers, α-blockers, calcium channel blockers, and spironolactone therapy. Model 3 adjusted for serum creatinine, hemoglobin, CRP, RDW, and albumin. *p* < 0.05 vs. RBCLS ≥ 70 d group. *p* for trend > 0.05.

## Data Availability

The data presented in this study are available upon request from the corresponding author.

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
