# Peer review of "Association of Red Blood Cell Life Span with Abnormal Changes in Cardiac Structure and Function in Non-Dialysis Patients with Chronic Kidney Disease Stages 3–5"

_jcm, 2022, doi:10.3390/jcm11247373_

Round 1

Reviewer 1 Report

Rao and colleagues have presented a provocative manuscript linking diminished RBC lifespan with cardiac function in non dialysis CKD patients.  They have demonstrated that significantly decreased RBC lifespan correlates with greater incidence of heart failure.

This is a very exciting area of research and highlights the often underappreciated role of diminished RBC lifespan in the anemia of renal disease.

This reviewer has a few suggestions to strengthen the work

1.  Utilization of Levitt's CO test to estimate RBC lifespan is clever and potentially applicable widely.  The authors describe the basis for the use of this test and the assumptions made that allow the authors to conclude that 70% of the expired CO derives from newly degraded RBCs.  However, an important consideration is whether or not this test has been validated in the CKD population.  This is a point that the authors should clarify and if this validation has not been performed in the CKD population, it should be listed as a limitation.  This reviewer could not find an explicit description of this validation in the literature.

2.  The authors describe their exclusion criteria somewhat briefly.  Specifically, did they check for hemoglobinopathy or formal pulmonary function testing?  Significant numbers of CKD patients may develop pulmonary HT or pulmonary vascular calcification.

3.  Inclusion of FGF23 and PTH levels would be of great interest and buttress the cardiac function data.  Likewise, systemic vascular or valvular calcification scores could also yield potential mechanisms for diminished RBC lifespan

4.  The authors may want to temper their conclusion that they have definitively demonstrated increased hemolysis until they have clarification on point 1.

Reviewer 2 Report

In the present study, the authors show in non-dialysis patients with CKD stages 3-5 that there is a correlation between red blood cell lifespan and cardiac struture and function. 

It has already been shown that survival of RBCs is progressively reduced in CKD patients. However, the potential impact of RBC life span on CV outcomes in CKD patients is quite interesting.

A main point in this study is related to the fact that there is no information on Epo therapy in the study population that includes mainly stage 5 CKD patients. In fact, Epo therapy has also been associated with cardiac struture and function in CKD and appears not to affect the lifespan of RBCs. Thus information on Epo therapy must be included in the data collected and the Epo therapy should be included in the adjusted analysis of the correlation between red blood cell lifespan and cardiac structure and function. 

Reviewer 3 Report

This is an interesting original article demonstrating an association of red blood cell life span (as measured by an improvement of the carbon monoxide breath test) with abnormal changes of cardiac structure and function in non-dialysis patients with CKD stages 3-5. ANy information regarding CKD patients and potential targets to improve their prognosis (regarding the CARDIOREANL syndrome) is absolutely welcome.

However, the main problem of this article is that they show an ASSOCIATION but their results do not demonstrate an INDEPENDENT effect. Many potential residual confounders have not been considered and therefore it can only be accepted if authors clearly underline the nature of their findings (just Association!...not CAUSALITY).  Multivariant analysis including many other factors should be evaluated (notice that many characteristics expressed in table 1 denote BASAL differences which could affect the independent effect of red blood cell lifespan). they should be considered in multivariant analysis (among others).

1) ABSTRACT: what does combined heart failure really means? Not clear for readers. Morover, I am not sure how heart failure is defined considering that BNP values are valid for the general population but probably not in CKD. (for example in Table 3)

2) MATERIALS and METHODS:

- Authors exclude patients on Roxadustat. What about treatment with Erythropoietin-stimulating agents, Oral and or iv Iron? It must be included anad analyzed. 

- What about patients with structural valvular disease?

- Please, explain the readers better about the "improvement" of the CO breath test. 

-Iron status should also be reflected as Transferrin Saturation Index (see table 1)

- Review last sentence of the point 2.3 ...missing verb?

-Why BNP level > 100 pg/ml was chosen in a CKD population? BNPO levels are clearly affected by CKD. 

- ADQI group recommends indexing left ventricular myocardial mass to CS or hight la L.S. Chawla, C.A. Herzog, M.R. Costanzo, J. Tumlin, J.A. Kellum, P.A. McCullough, et al.Proposal for a functional classification system of heart failure in patients with end-stage renal disease: proceedings of the acute dialysis quality initiative (ADQI) XI workgroup.J Am Coll Cardiol, 63 (2014), pp. 1246-1252. Please comment. 

-Please, notice that  ADQI group and ESC use of LA y>34mL/m2. Burkert Pieske, Carsten Tschöpe,et al. How to diagnose heart failure with preserved ejection fraction: the HFA–PEFF diagnostic algorithm: a consensus recommendation from the Heart Failure Association (HFA) of the European Society of Cardiology (ESC), European Heart Journal, Volume 40, Issue 40, 21 October 2019, Pages 3297–3317, https://doi.org/10.1093/eurheartj/ehz641

3) RESULTS 

- What do the black points mean in Figure 1?

- Table 1: As mentioned before, notice all the significant BASAL differences among the different gropus. they should be considered in the adjustament for confounders.. Serum ferritin is much much greater in aptients with the lowest RBCL and this is probably due to iron treatment. Differences in treatments (for cardiac pathology, hypertension, Ca-P-PTH, anemia should also be analyzed). Do you have FGF-23  values? Inflammation parameters such as CRP?

- Please, underline in the abstract and discussion that most of the sample is CKD stage 5. Why RBCL isnt lower in this more severe group? Moreover, in Table 1 it is shown that there are no significant differences in RBCL and GFR. Shouldnt it go against CAUSALITY?

4- DISCUSSION

Why right ventricle parameters are not included. Be aware that its importance is absolutely increasing 

- The hypothesis (such as the  of arginase activation) makes no sense if it is not proven first the INDEPENDENT effect 

Congratulations for the excellent initiative! 

Round 2

Reviewer 2 Report

The manuscript can now be accepted in its present form

Reviewer 3 Report

Thanks for giving xplanations to my queries